# Spatial immunization to abate disease spreading in transportation hubs

Mattia Mazzoli [1,2] ✉, Riccardo Gallotti [3], Filippo Privitera[4], Pere Colet [1] & José J. Ramasco [1] ✉

Proximity social interactions are crucial for infectious diseases transmission. Crowded agglomerations pose serious risk of triggering superspreading events. Locations like transportation hubs (airports and stations) are designed to optimize logistic efficiency, not to reduce crowding, and are characterized by a constant in and out flow of people. Here, we analyze the paradigmatic example of London Heathrow, one of the busiest European airports. Thanks to a dataset of anonymized individuals' trajectories, we can model the spreading of different diseases to localize the contagion hotspots and to propose a spatial immunization policy targeting them to reduce disease spreading risk. We also detect the most vulnerable destinations to contagions produced at the airport and quantify the benefits of the spatial immunization technique to prevent regional and global disease diffusion. This method is immediately generalizable to train, metro and bus stations and to other facilities such as commercial or convention centers.

People density, and its translation into contacts, is a crucial factor in the propagation of contact diseases[1–3]. Although hygienic measures can decrease contagion probabilities[4], airborne diseases pose a special challenge since they spread via droplets and aerosols, affecting the local environment, including surfaces where they deposit and the pathogen can continue to be infectious for a certain time period[5]. While crowding might be averted in occasions such as demonstrations, sport events, theaters, etc, it is much harder to avoid in public transportation networks. These systems have been designed with transportation efficiency in mind and crowding, especially in large hubs, is an inherent consequence. There are several works dealing with contagion events on buses, cruise ships[6], trains[7–11], and airplanes[12–15], which in general depend on the infectivity of the disease, the ventilation system, the duration of the trip and the occupation of the vehicle. Beyond the proposal of screening systems such as fever (temperature) detectors[16–20], much less attention has been devoted to infectious events in transportation hubs themselves as in airports.

The fast spread of viral diseases worldwide is due to long-range aerial trips connecting countries everyday[21–24]. For this reason airports are very sensitive places[25,26], where not only infected passengers can depart and seed their final destination regions, but they can also infect other passengers along the terminal corridors going to many other destinations and, thus, amplifying both the geographical variety of the epidemic and its spreading velocity. Promptly controlling contagion events in airports can lead to a non-negligible effect on the global containment of the virus[19,20,26,27], even though as shown in the literature[28–32] travel bans (if not issued on a wide scale) only lead to delays since alternative pathways are available for the spreading.

In order to study this process at an individual scale, where the unit of measurement is human contact, we need high-resolution proximity sensors to detect human movements in public spaces in detail. This allows us to better study human mobility patterns in everyday life environments and understand where to strategically intervene. Leading the research in this sector, the SocioPatterns project allowed many recent studies by using wearable RFID proximity sensors in different contexts[33–37] as scientific conferences[38], hospitals[39,40], offices[41], museums[42] and schools[43–45]. The most important issue addressed by these works is the study of human contact networks, which has been analyzed from many points of view and for different purposes, like data reconstruction, data cleaning and sampling biases[46–54]. This

[1]Instituto de Física Interdisciplinar y Sistemas Complejos IFISC (CSIC-UIB), Campus UIB, 07122 Palma de Mallorca, Spain. [2]INSERM, Sorbonne Université, Institut Pierre Louis d'Epidémiologie et de Santé Publique, IPLESP, Paris, France. [3]CHuB Lab, Fondazione Bruno Kessler, Via Sommarive 18, 38123 Povo (TN), Trento, Italy. [4]Cuebiq Inc., 45 W 27th Street, 3rd floor, 10001 New York, NY, USA. ✉e-mail: mattia.mfn@gmail.com; jramasco@ifisc.uib-csic.es

branch of network science led to answer open questions such as the definition of contacts and especially the effect of using different types of networks to mimic human contacts in epidemic models[55,56].

In this work, we use GPS mobility data of anonymized and opted-in individuals in Heathrow Airport, London UK, to build a contact network based on copresence in $10 \times 10$ m$^2$ cells. Over this network, we run compartmental epidemic models representing different diseases, which allows us to assess the risk of local contagions of passengers with intercontinental, European, UK destinations and those arriving in the city of London. The models are intended to explore the effect of spatial immunization in which policies or devices are implemented to reduce the probability of contagion in certain cells, an example of which is the use of UV-C lights. Our spatial immunization is intended to run as a background silent policy, whose importance comes at play at the moment of having completely undetected first imported cases. This is why in our models we do not introduce any other intervention protocols, which typically are introduced once the outbreak is identified. The most effective areas to place these devices coincide with the locations with the largest number of contagions. Interestingly, these places can be already identified with a simple SIR (Susceptible - Infected - Recovered) model and they are not necessarily those through which pass the largest number of individuals, since the probability of contagion also depends on the type of individuals (workers versus passengers to/from different destinations/origins) and the time they spend in the cell. We quantify the effects of different coverage levels in terms of the amount of space immunized on the spread of the diseases. Our methodology is general, since the same technique can be applied to any public building.

## Results

### Copresence dynamic network

Trajectories in the airport of six moths are supersampled to generate a standard day (see Methods). A contact network for each 15 mins time slot is built based on copresence patterns of every pair of individuals: a link is established if they coincide in the same spatial cell at a given

time slot. A sketch illustrating the process is displayed in Fig. 1a. A network is built for each time slot, and the set of networks is used in the epidemic spreading simulations. To have a first glimpse on the structure, Fig. 1b shows the result of aggregating the copresence networks over a month. The colors represent the type of individuals: passengers in red and workers in blue. Note that on this aggregated network connections can be multiple (links are weighted) if individuals exhibit cell copresence at more than one-time slot. We also display the aggregated subnetworks between passengers (Fig. 1c) and workers (Fig. 1d) alone. The subnetwork of workers is highly connected since they spend long periods of time at the airport, whereas the same cannot be said for travelers. In fact, workers sustain the connectivity of the aggregated contact network. On the other hand, passengers just come and go continuously, making contacts with some workers at the control, commercial areas and may coincide with other passengers sporadically at duty-free zones, bars and restaurants and boarding gates.

The copresence dynamic network is used as the skeleton to run the epidemic models. These models are based on a compartmental approach, in which individuals have a variable of status associated to the disease (susceptible S, infected I, exposed E, recovered R, etc) and they transition from one to the other following the disease clinical progression. The models have been adapted to several known airborne diseases as illustrative examples: influenza (SEIR model), SARS (SIR) and COVID-19 (SEIIR model). However, as shown later, our methods are general enough to cover multiple diseases. In fact, we consider the scenario of the arrival of an undetected pathogen and how spatial immunization techniques can help to mitigate its global propagation. Details on the building of the models and their parameters are provided in the Methods section below.

### Spatial immunization

We run a SIR epidemic model on the temporal co-presence network (see Methods below for details in the model). Infection events are registered at the level of cells for each realization. This allows us to classify the areas according to the risk of developing contagions in them. Figure 2 shows an example obtained for the first two days of the SIR model simulations. The heatmap shows the density of cumulative contagions occurring after one and two days in each cell. One can observe the main terminals in the map: terminal 5 and satellites on the left with their access road in the top of the image, terminals 1-2-3 in the center and terminal 4 on the bottom. Terminal 1 was closed in 2015, and terminal 2 was expanded over it. We keep the terminology terminal 1-2-3 to refer to them. In the terminals, it is possible to identify the fraction of flights linked to specific destinations (intercontinental, UK or European). There are some areas whose access is restricted to airport staff in the terminals, across the runaways, on the bottom left, and on the right-side and top of the maps. As the time goes by, the picture in terms of the density of contagions in Fig. 2 becomes sharper, with more peripheral cells in the terminals starting to show more contagion events. This is important because these cells are the obvious candidates to be immunized to hinder disease spreading. Interestingly, such cells are not necessarily the most crowded ones. The probability of contagion is driven by a balance between the number of people present in a place and the time that they spend there. In the interior of the terminals, some of these cells match with security control, service and retailing zones.

We rank cells by number of contagion events on the first day, and select a certain number of them from top to down to be immunized. The impact of an increasing number of immunized cells over the disease spreading is the main focus of Fig. 3. Given that the infectivity per contact $p_\beta$ is sensibly smaller in treated areas, the number of realizations that display contagion events in the airport, $p_R$, decreases as more cells are immunized (see Fig. 3a). Already with 400 immunized cells (1.1% of all available cells) the number of realizations with

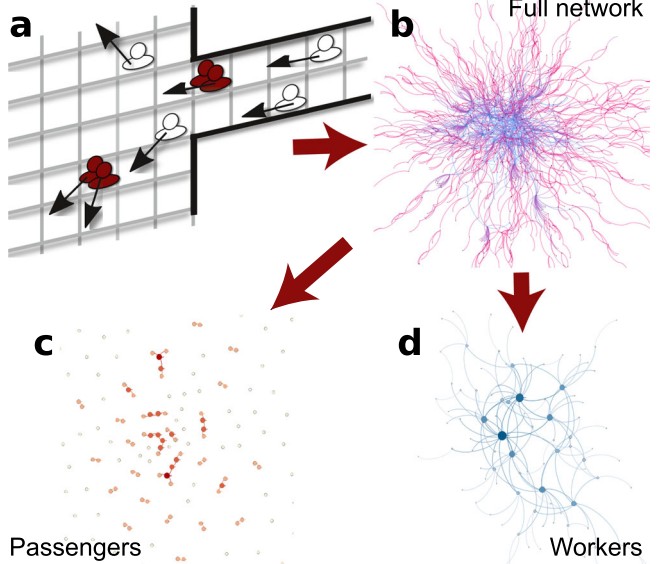

**Fig. 1 | London Heathrow Airport aggregated copresence network of one-month data. a** Sketch of the contact network construction: two individuals are linked if they coincide in a cell within a time slot. **b** Cumulative copresence network of 1 month. In blue the workers, in red the passengers. User type assortativity $a_i = 0.45$. **c** Travelers copresence network is a *fragmented* graph since there is an incoming and outgoing flow and they spend a relatively short period in the terminals. **d** Workers copresence network is a connected graph: they interact recurrently with each other.

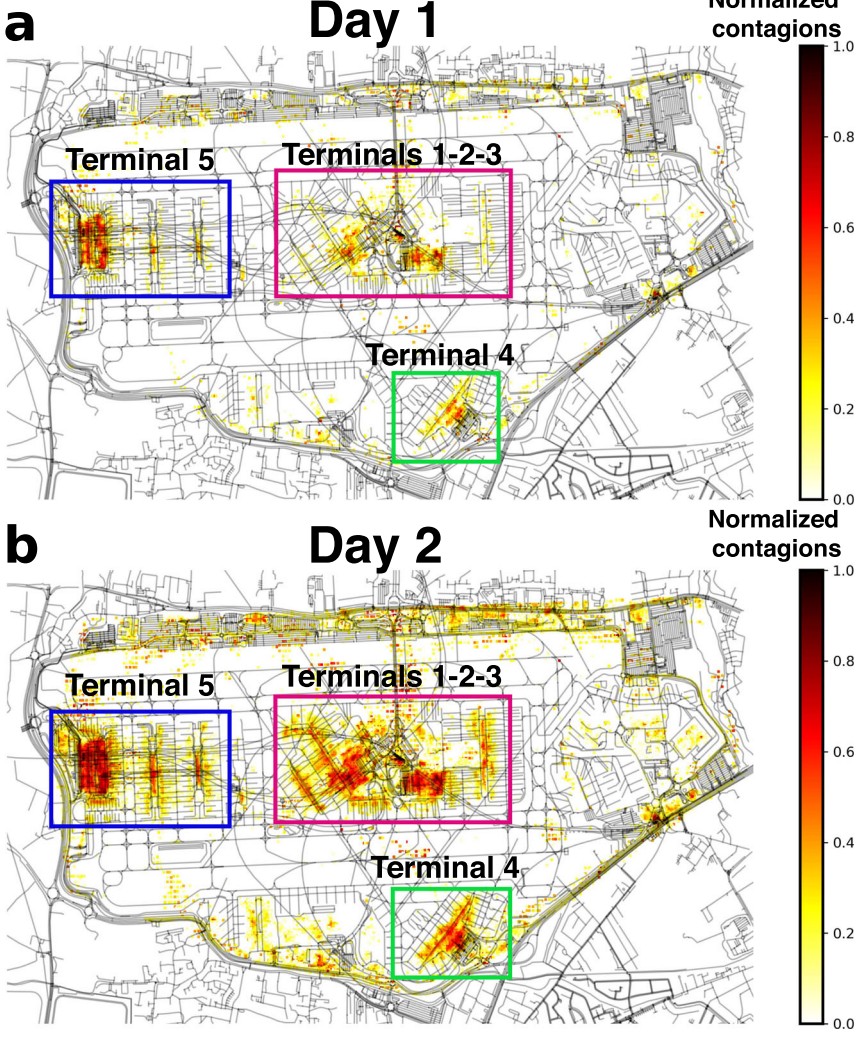

**Fig. 2 | Density of contagions in space.** Heatmap showing the cells in the airport area in which contagions occur when running the SIR model. In **a**, after one day of simulation and in **b** after two days. Color code normalized for each period considered.

outbreaks is reduced by 20% of the one without immunization. The effect of the immunization on the amount of initial infections can also be seen in the boxplot with the number of contagions on the first 24 hours since the arrival of the first case as shown in Fig. 3b.

Additionally, for realizations with outbreaks, the time evolution of contagions at the airport facilities changes notably as well. Contagions of the second and later days are mainly due to infected workers who return to work everyday. While this may not be a fully realistic assumption for diseases as SARS, in which infected individuals normally develop severe symptoms, the parameters used in the SIR model are quite similar to those for other coronavirus-induced diseases such as common colds that produce mild symptoms and would allow infected individuals to keep working. Figure 3c shows the number of infections averaged over realizations with outbreaks as a function of time for different numbers of immunized cells. The light blue line corresponds to the case without spatial immunization and is taken as baseline. Infections occur mostly during the day, when most passengers and workers are present at the airport. The largest peak takes place on the third and fourth days resulting from the workers who have been infected on the first day. This peak is over 4 times higher than that of the first day. Infection rates remain high on subsequent days.

The introduction of spatial immunization has two relevant effects on the evolution of contagions during the first week: On one side, it lowers the curve of new infections for all days. On the other, it delays

the peak of contagions, providing time, for instance, for contact tracing once the outbreak is detected. These two effects can be seen already with 100 immunized cells (green line in Fig. 3c), where the contagions peak takes place now on the fourth day and is even more manifest for 400 (1.1%) cells for which most of the contagions on the initial two days are prevented, the contagion peak is delayed practically until the last day and its height is smaller than half that of the baseline case. Increasing the number of cells leads to further lower the infections curve.

We now consider the overall number of infections over the 7 days period of analysis, but dis-aggregating affected individuals in five categories: workers, passengers arriving to London, passengers traveling to UK, EU and other international destinations. Figure 3d shows for each of these categories the number of infected individuals $N_i$ when implementing $i$ immunizing cells normalized to the baseline case without spatial immunization $N_0$. Reduction of contagions takes place systematically for passengers to any destination as well as for workers, albeit for this last ones is less marked. Finally, in Fig. 3e we plot the distribution of infected individuals by category normalized to the total number of infections in each configuration, namely $i_i^j = N_i^j / \sum_k N_i^k$ where index $j, k$ refer to the 5 categories considered and $i$ to the number of immunized cells. This last plot confirms what we already saw in Fig. 3d, where workers and, to less extent, intercontinental passengers are less protected than others by the spatial immunization,

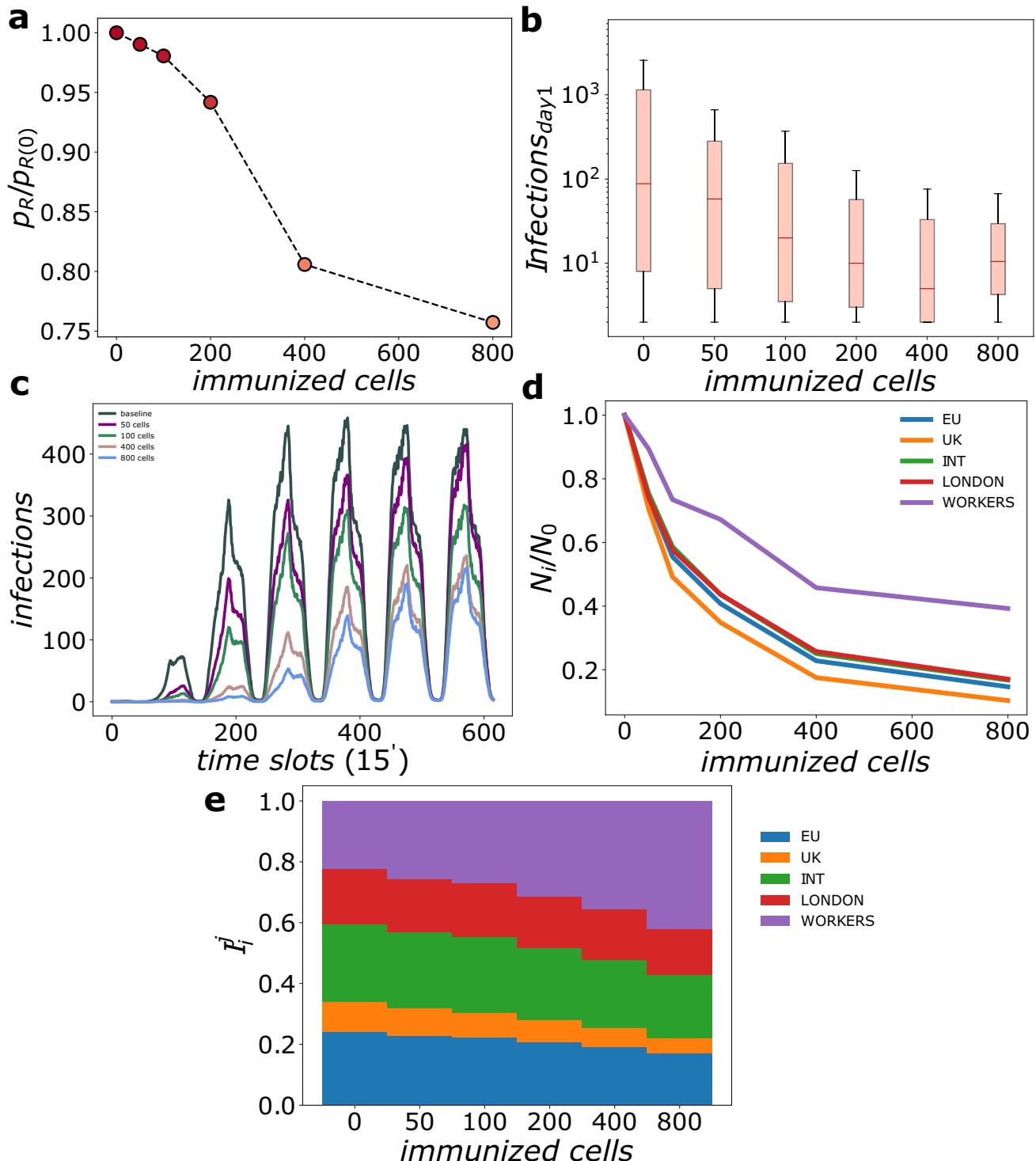

**Fig. 3 | Spatial immunization effect on SIR propagation. a** Number of realizations with secondary infections in the airport facilities $p_R$ as a function of the number of immunized cells, normalized to the baseline case without spatial immunization $p_R(0)$ (each cell extension is 100 m²). **b** Boxplot of the distribution of contagions in the first 24 hours. Medians are shown as red lines, boxes represent the IQR, whiskers extend to 1.5 × IQR above and below the 75 and 25 percentile respectively. **c** New infections per time slot as a function of time and for different spatial immunization configurations. **d** Infections over the 7 day simulation period disaggregated in 5 categories: workers, passengers arriving in London, traveling to the UK destinations, to EU destinations and to other international destinations. For each category, $N_i/N_0$ stands for the ratio between the number of infections with $i$ cells immunized over the baseline without spatial immunization. **e** Relative composition of the infected population (see text). For each scenario we run $n = 1000$ simulations. See Methods and Table 1 for acronyms definitions.

while UK passengers are the least affected even with no policies implemented.

As a robustness check, we have run simulations with this basic SIR model changing the time period considered to build the contact

network, hence with a different definition of contact. Longer times naturally lead to more contacts since it is easier for people passing by an area to interact with other individuals who have been present there longer before. Instead of 15 minutes, we have built the contact network

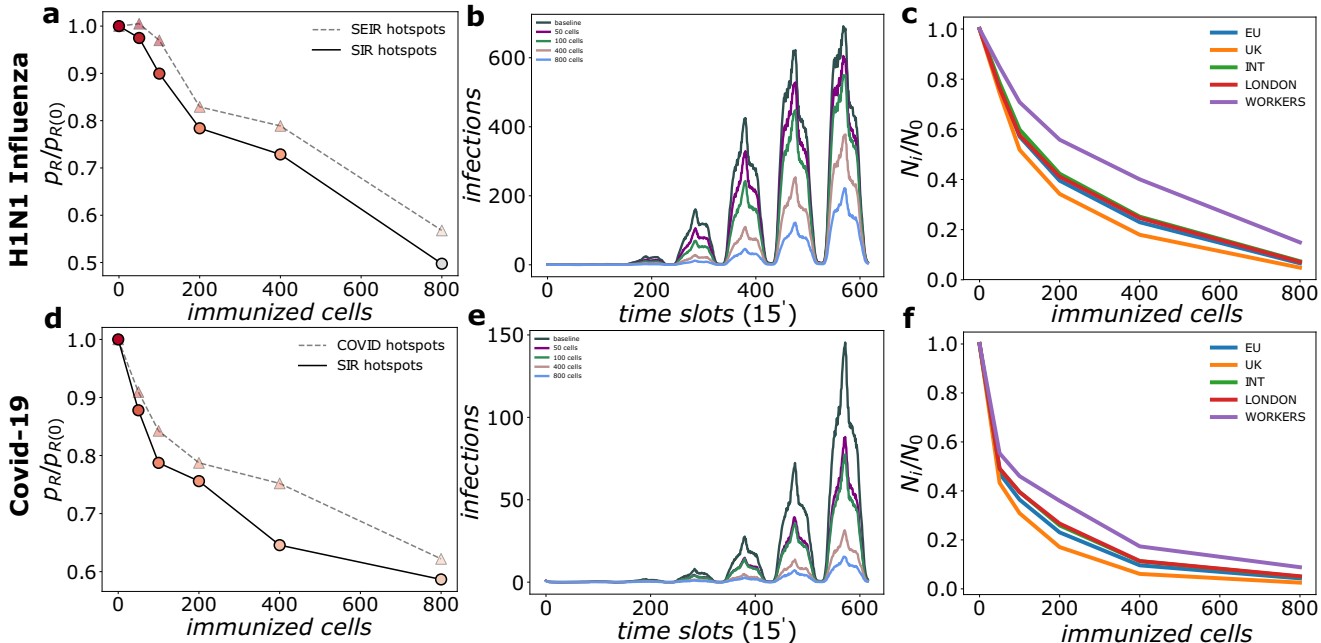

**Fig. 4 | Spatial immunization applied to the spreading of diseases described by SEIR models.** Panels **a**, **b** and **c** refer to results of simulations for H1N1 Influenza; And panels **d**, **e** and **f** to those for COVID-19. In terms of columns, panels **a** and **d** show the number of realizations with secondary contagions $p_R$ normalized by $p_R(0)$ (without spatial immunization) as function of the number of immunized cells. The cells to immunize are selected following the contagion ranking according to the i) SIR model (circles joined by solid black lines) and ii) corresponding SEIR model ranking (circles joined by grey dashed lines). Panels **b** and **e** show the average number of new infections per time slot. Panels **c** and **f**, infections over the full simulation period disaggregated by workers and passenger destinations. For panels **b**, **c**, **e** and **f** immunized cells have been selected according to SIR ranking. For each scenario we run $n = 1000$ simulations. See Methods and Table 1 for acronyms definitions.

taking the double of time, 30 minutes, as a basic time unit. The results are shown in Supplementary Figs. S10 and S11. The number of contacts and of infections increase, but the dynamics of new infections shows a similar pattern as the 15 min definition of contact (Supplementary Fig. S10) and, most importantly, the cells where most of the contagions occur are highly correlated (Supplementary Fig. S11).

**Spatial immunization performance**

In order to prove the performance of this policy when infected individuals undergo a latency period of the disease, hence for diseases with longer generation times, we run simulations for the SEIR family models taking as reference the cases of H1N1 Influenza and COVID-19. As for the SIR model, we first discuss the early stages of the spreading process since passengers, the ones prone to spread the disease worldwide, spend short periods of time in the airport. For the first day after the first case arrival, SIR and SEIR models yield similar results in terms of where the contagions occur. Thus, the ranking of the cells obtained from the SIR model with the parameters considered in the previous section can be taken as a good proxy to determine contagion hotspots to be immunized regardless of the specific disease. This can be observed in Fig. 4a, where the number of contagions on the first-day $p_R$ obtained for the SEIR model with the H1N1 Influenza parameters is displayed as function of the number of immunized cells taken accordingly to the SIR ranking (circles joined by solid black lines) and the SEIR ranking (circles joined by grey dashed lines). The results are almost equivalent and, being simpler, from a computational perspective, SIR simulations are more efficient for the estimation of the contagion hotspots. In any case, spatial immunization systematically reduces infection spreading: already 400 (1.1%) immunized cells reduce the number of realizations with H1N1 secondary contagions by 30%.

We now focus on the spreading dynamics over a week. As for the SIR model infections occur mostly during the day and infections

occurring beyond the first day are mainly sustained by workers. At a difference for the SIR model, for the baseline case without spatial immunization the slower progress of the disease brought by the introduction of the exposed phase delays the peak of contagions until the last day as shown in Fig. 4b. Introduction of spatial immunization lowers the curves and delays them, albeit this last effect is now much less noticeable since the contagion curve was already delayed by the exposed phase.

Finally, Fig. 4c shows the infected individuals over the 7 days period of the simulation disaggregated in the same categories as before: workers, passengers arriving at London, traveling to UK destinations, traveling to EU destinations and passengers traveling to other international destinations. Spatial immunization reduces contagions systematically for all categories although to a lesser degree for workers. See Supplementary Figs. S12-S14 for sensitivity analyses including lower immunization efficacy and slightly altered mobility.

In the case of a COVID-19 importation in the airport, in Fig. 4d, we plot the number of realizations with local contagions in the airport as function of the number of immunized cells where, as for the previous cases, the cells to be immunized has been determined using the ranking from the SIR model (circles joined by solid black lines) and from the SEIIR model (circles joined by dashed gray lines). The SIR model is a convenient and efficient proxy to rank the cells to be immunized. Looking at the effect of immunizing a progressive number of cells, a 35% reduction in the number of realizations with secondary infections can be achieved with 400 (1.1%) cells.

Considering now the evolution over a week, Fig. 4e shows that the introduction of spatial immunization reduces the number of contagions per time slot by a factor of 3 already with 400 (1.1%) cells. Further reductions can be achieved with a larger number of immunized cells.

Similarly to the other diseases discussed before, workers are the most infecting category, as Fig. 4f shows and they are the least affected

by the implemented policy, which seems to have a comparable effect on all the other categories. Due to the observed increase of transmissibility of new COVID-19 variants and a lack of a clear estimation of the wild-type reproductive number, we reproduced our simulations with other values of $R_0$ in Supplementary Figure S15. The plot shows how the immunization policy is still effective in the reduction of outbreak intensity.

## Discussion

In this work, we have shown how to reduce the complexity of the spatial trajectories of individuals in Heathrow airport by encoding them in a temporal contact network generated by copresences in small cells among different categories of individuals. The trajectories followed by passengers and workers are extracted from GPS mobility data of application records over a 6 months period and we supersampled them creating a synthetic day with a number of trajectories similar to that of a standard day at the airport, which in our simulations we repeat multiple subsequent times. A key ingredient in our methodology is the classification of individuals in passengers, present at the airport for a much shorter time than that characteristic of epidemics, and workers, returning to the airport daily. On top of the synthetic copresence network, we consider three different compartment models from the literature to analyze the spreading of SARS, influenza and COVID-19. Numerical simulations show that workers play a key role in the spreading dynamics independently of the virus under study. This is due to the fact that employees are the agents more often infected: they recurrently return to the airport day after day and they come in contact with many passengers and with their colleagues, whereas passengers generally remain at the airport only for a few hours before departure or after arrival. Second in this rank by infectiousness, there are connecting passengers since they are the travelers who spend more time inside the airport. Restaurants, bars and in general relax areas are the places where most of the copresences (and contagions) happen and this is where indeed connecting travelers use to rest and stop for a long time waiting and interacting with airport workers.

The application of spatial immunization policies based on methods such as the use of non-harmful UV lights, frequent cleaning and disinfection of surfaces, air filters, ozone, etc, to hygienize specific terminal areas sensibly reduces both the number of realizations with secondary contagions at the airport and the intensity of the local outbreaks. Contagions within the airport facilities taking place at the very initial stages are key for a world-wide spreading of diseases. Contagion reduction can be efficiently achieved by prioritizing the areas to immunize according to the infection hotspots predicted by a SIR epidemic model. The SIR model, despite its simplicity, seems to be enough to capture the essence of transmission dynamics. We have shown that even this model, which does not necessarily suits a real disease progression, can be used to design targeted interventions in case of first importations of various diseases. This result is particularly relevant in practice, since it implies that a single hotspot configuration is very effective for many diseases regardless of the disease-specific epidemiological parameters. Note that these hotspots are not only characterized by the density of people but also by the combination of time spent, recurrence of contacts and individuals density: places as cafeterias, access points, etc, are the most obvious areas to immunize, although our method is able to capture less obvious locations as some particular gates, shops or locations representing bottlenecks in the airport mobility. Given the uncertainty in the transmission probability of an emerging disease, we have checked the robustness of our spatial immunization policy to reduce contagions by varying the transmissibility of a disease (see Supplementary Sec. Generalizability of the method in the SI).

By further disaggregating individuals into workers, passengers arriving to London and those traveling to UK, European and intercontinental destinations, we estimate the average threat represented by the development of an outbreak in the airport for different destinations. We find an important decrease in the amount of infections for all categories of individuals when spatial immunization is implemented. Therefore, this method is helpful in containing the menace of and potentially delaying the seeding of emerging diseases from an airport hub to the rest of the world.

This method seems to be less effective on workers, who exhibit a longer exposure to the virus with respect to passengers. This suggests that for them it becomes necessary to complement spatial immunization with other procedures, such as targeted vaccinations in order to reduce the probability and even the intensity of an outbreak within every scenario. Still, in the early pandemic stages, when vaccination is not available, spatial immunization is an important tool to decrease the risk of global propagation.

Finally, we would like to remark that the methodology presented here can be directly applied to other communication hubs, including train, metro and bus stations as well as to crowded facilities such as commercial centers, department stores or convention centers in order to reduce the probability of new and uncontrolled outbreaks. All these places share the common feature of having a large in and outflow of visitors and a set of workers that are recurrently present in specific areas of the infrastructure. This leads to strong heterogeneities in the location and duration of the contacts among visitors and between visitors and workers, which allows us to define a successful spatial immunization strategy based on a hotspot analysis.

## Methods

### Building the copresence dynamic network

A first question to address is how to determine the copresence contact structure. To this end, we use GPS mobility data from smartphone application records collected by Cuebiq on anonymized individuals who have opted-in to the service through an European Union General Data Protection Regulation (GDPR) compliant framework. The data includes individuals' trajectories inside the airport from February to August 2017. In order to preserve the privacy of individuals living in the neighboring areas, we do not analyze information outside of the airport perimeter. Also, users with less than 10 location points in the dataset are discarded. The number of individuals recorded inside the airport on an average day in the dataset is not sufficient to reproduce a realistic traffic scenario, hence we have to supersample our records. In order to do so, we consider all trajectories observed in the 6-months period as taking place in a single day. This allows us to build a synthetic population of 206,043 individuals, including 143,588 passengers and 62,639 workers, a size similar to that of the standard day at Heathrow airport[57] (see Supplementary Information Subsec. Supersampling for details). This *standard* day covers trajectories of 24 hours, without losing continuity on night-time trajectories, to take into account individuals (mostly workers) spending the night at the airport facilities.

Given that the mobility data is not continuously collected, some individuals are observed consecutively in distant cells, hence we do not always know the precise actual path. This lack of information would produce a sensible subestimation of contacts in the airport, hence we need to apply a trajectory reconstruction process.

As a first step, we discretize time and space to facilitate the computation, dividing the airport in small cells of $10 \times 10$ m² and considering time slots of 15 minutes. These values are a balance between having sufficient fine information within the terminals and the dataset resolution limitations (spatial and statistical) that may introduce artificial noise[58]. We filter out all those cells with a number of visitors below a given threshold, under the assumption that they are associated with nonaccessible areas. These cells main not used in the trajectory reconstruction. Although the main results of the paper are for a threshold of 30 visits, we have

checked that similar results are obtained using different thresholds: 10, 20, 30, 40 and 50 visits per cell (see Supplementary Information Subsec. Trajectories reconstruction and Supplementary Figs. S1-S2). Users' trajectories are completed with cells belonging to the shortest path, compatible with the infrastructure geometry, between two consecutive observation points, provided that they are not farther than 50 m. By this process, we reconstruct approximately 4 millions intermediate location points, finally reaching a total of 10 millions data points.

Users are classified as passengers and workers according to the spatio-temporal patterns of their presence. The users identified as workers include staff of the airport, the airlines or the stores and restaurants placed across the terminals. Essentially, to identify workers we are capturing features that are unlikely for travellers: workers are defined as individuals who are either observed in the airport for three or more consecutive days with long visits (>4.5 h) or those entering staff-only permitted areas out of the terminals. On the other hand, for passengers, we can have a hint of their destination/origin using a semantic analysis of their trajectories. If they start at the entrance, metro or bus area, and end at the gates, they are departing passengers. The other way around, trajectories going from gates to exit identify arrival passengers. And, finally, if the initial and final points are at the gates they are connecting passengers (see Supplementary Information Subsec. Travelers classification, Supplementary Figure S3). Furthermore, by using the destinations of flights departing from each terminal[59] it is possible to assess the risk of transmission to other UK cities, to Europe or to intercontinental destinations.

Once the users are divided in the two main categories of workers and passengers, the "standard" day can be repeated to simulate longer time periods. We consider that users associated with workers remain the same through the simulation period, while those associated with passengers are renewed every day.

## Epidemic models

The temporal contact network informs the epidemic models for simulating the contagion process independently from the disease considered, although every disease exhibits its own epidemiological parameters and clinical progression. We adopt a compartmental framework where individuals assume different states (compartments) representing the disease progression. The simplest model of this family is the so-called SIR, in which individuals can be Susceptible to the disease, Infected-Infectious, or Recovered-Removed (which is equivalent to being immune). The model dynamics in a continuous time context can be represented as follows

$$S + I \xrightarrow{\beta_t} 2I,$$
$$I \xrightarrow{\mu} R. \tag{1}$$

The first reaction equation states that when in contact, an infectious individual $I$ can infect a susceptible one $S$ with a transmission rate $\beta_t$. In the simulations, where the time is discretized to 15 min slots, this process is implemented with a probability $p_\beta$ of contagion per contact. The second equation captures how an infectious individual $I$ can recover to become $R$ with a probability rate $\mu$, where $\mu^{-1}$ is the characteristic recovery time. While constant rates are a good approximation to a population level, at individual level we need to take into account heterogeneity in the infectious period and, therefore, we use Gamma distributed recovery times $t_r$ with average $\mu^{-1}$[60]. Formally, for each infected individual we extract $t_r = \Gamma(\mu^{-1}, 1)$ from a Gamma distribution with shape $\mu^{-1}$ and scale 1. In this model, the interplay between these two parameters and the contact network is captured by one important control parameter that is the Reproductive Number $R_0$ (see Supplementary Section Modeling epidemics in open systems for details on the parametrization of the model). More intricate models can be easily built

under this framework as, for instance, the SEIR model, in which the new infected individuals go through a latent or exposed $E$ phase before becoming infectious. The reaction equations are

$$S + I \xrightarrow{\beta_t} I + E,$$
$$E \xrightarrow{\gamma} I, \tag{2}$$
$$I \xrightarrow{\mu} R,$$

where $\beta_t$ and $\mu$ play the same role as before, and $\gamma$ is the probability rate at which exposed individuals become infectious.

Depending on the values of the parameters, these models can represent the spreading of different diseases. For instance, SIR has been used to simulate SARS propagation[61], while slightly more elaborated versions of SEIR have been used for influenza[62] and COVID-19[63]. Table 1 lists the models employed in this work and the references from which we have obtained the estimations for the parameter values. It is important to mention here that, since we have a continuous flow of people, we are mostly interested in the contagions at the early stages of an outbreak. These are the ones that can propagate the disease in an airport and from there to other world destinations, given that further development of the disease would lead to airport closure. Further details on each model are provided below on a case by case basis. In Table 2 we list the main parameters for the models implemented in this study.

**Simplest model: SIR.** Based on previous works modeling and analysing the SARS (SARS-CoV-1) outbreak of 2002-2004[61,64], we inform an SIR (Susceptible - Infected - Recovered) model with a Gamma distributed infectious period and mean $\mu^{-1} = 10.6$ days (see Eq. (1)). We also fix the probability $p_\beta = 0.92 \times 10^{-3}$ of infection per contact, so that we would recover $R_0 = 2.7$ with our rate of contact per time unit for a well-mixed system (estimated for SARS in ref. [64]) see Supplementary Subsec. Modeling epidemic spreading in open systems, Supplementary Figs. S4-S6. As a way to perform a risk analysis, the model is run over 1000 stochastic realizations. In our simulations, the first infected individual (seed) lands at the airport at 13:30 local time on the first day. The origin of the flight, as the corresponding arrival gate, are selected

## Table 1 | List of abbreviation for the epidemic models here considered and respective reference to the literature for the choice of the epidemic parameters

| Name | Explanation | Reference |
|------|-------------|-----------|
| SIR | SIR model for SARS-CoV-1 | [61,64] |
| SIRR3 | SIR double infectivity | [61,64] |
| SIR30 | SIR with 30' time scale | [61,64] |
| SEIR | SEIR for H1N1 | [62] |
| COVID-19 | SEIIR for SARS-CoV-2 | [63] |

SIR stands for Susceptible - Infected - Recovered, SEIR stands for Susceptible - Exposed -Infected - Recovered, SEIIR stands for Susceptible - Exposed -Infected - Recovered where the Infected compartment is split in asymptomatic and symptomatic.

## Table 2 | List of parameters for the epidemic models here considered

| Name | $p_\beta$ | $\mu^{-1}(d)$ | $\epsilon^{-1}(d)$ | $\mu_p^{-1}(d)$ |
|------|-----------|---------------|---------------------|------------------|
| SIR (Sars) | $0.92 \times 10^{-3}$ | 10.6 | – | – |
| SEIR (H1N1 Flu) | $3.06 \times 10^{-3}$ | 2.5 | 1.1 | – |
| SEIIR (COVID-19) | $4.31 \times 10^{-3}$ | 2.3 | 3.7 | 1.5 |

$d$ stands for days, $p_\beta$ is the probability of infection per contact, $\mu^{-1}$ is the infecitous period, $\epsilon^{-1}$ the latency period, $\mu_p^{-1}$ is the prodromic period.

at random between the options available. Then the simulation continues for seven consecutive days (clones, in terms of contact sequence, of the *standard* day, as discussed above). The presence of the seed in the airport may or may not lead to the emergence of contagions.

**SEIR family.** The next model considered is a little more involved, since now the individuals after contagion undergo a latent phase (E for Exposed) before becoming infectious. Such SEIR (Susceptible - Exposed - Infected - Recovered) frameworks have been employed to model the evolution of a number of diseases as, for instance, influenza and COVID-19.

**H1N1 Influenza.** To have another concrete example, we focus on the case of the H1N1 Influenza that was behind the 2009 pandemic. The model parameters are recovered from[62], which with our contact rates translates into an infection probability $p_\beta = 3.06 \times 10^{-3}$ per contact. Once infected, individuals enter in the exposed phase with an average duration of $\varepsilon^{-1} = 1.1$ days and, after, in an infectious period of $\mu^{-1} = 2.5$ days on average. In this phase, individuals have a probability $p_a = 0.33$ of being asymptomatic, a situation in which they are infectious but with an infectivity rate re-scaled by a factor $r_\beta = 0.5$. The rest of exposed individuals will become symptomatic infected. These parameters correspond to a basic reproduction number $R_0 = 1.75$[23] with our contact rate (see Supplementary Subsec. Modeling epidemic spread in open systems, Supplementary Figs. S4-S6). The simulation setting is the same as for the SIR model, with a single seed arriving at 13:30 the first day.

**SEIIR for COVID-19.** Finally, we analyze the effect of spatial immunization on SARS-Cov-2 spreading. We consider a model based on that proposed in a recent study[63]. This is a SEIR model where the I (infected) compartment is split in two: $I_p$ (prodromic infected) and $I$ (infected), which is further split into $I_a$ (asymptomatic), $I_m$ (mild symptomatic) and $I_s$ (severe symptomatic). The model of[63] includes also several compartments regarding hospitalization and ICU treatments, states that are severe enough as to be incompatible with being in an airport. We merge all these compartments in a single $R$ state. We have no specific information on the age of individuals, hence we treat them as the literature model treats adults. The model parameters are as follows: probability of infection per contact, $p_\beta = 4.31 \times 10^{-3}$, which leads to a reproduction rate $R_0 = 2.3$[65], the mean latency period of exposure $\varepsilon^{-1} = 3.7$ days, the average prodromic period before recovering $\mu_p^{-1} = 1.5$ days, and the mean infectious period before becoming recovered is $\mu^{-1} = 2.3$ days.

**Spatial immunization implementation**
The mentioned diseases have as common feature a contact (airborne) spreading. The main assumption in the models is that people staying in the same cell with one or several infected individuals have a certain probability of contagion. The idea behind spatial immunization is to reduce such probability in the most vulnerable cells. There are different non-exclusive methods leading to an effective reduction of the contagion probability, just to name a few: frequent cleaning and disinfection of surfaces, air filters, ozone and ultraviolet UV-C lights.

The use of ultraviolet (UV) light to sanitize spaces, for example, was first theorized in 1944[66] and experimented in 1981 in the Soviet Union by irradiating with UV lights of $\lambda = 254nm$ wavelength samples of various influenza viruses to study their photosensitivity. However, long human exposition to this spectrum of UV rays can be harmful. Hence, new studies proved the photosensitivity of airborne viruses to non-harmful UV lights in-vitro[67] and in-vivo[68], this time with a wavelength of the range $\lambda = 207 - 222nm$ (see, for instance[12], for a review and[69,70] for a

discussion on the use of UV-C to disinfect in the context of the SARS-CoV-2 pandemic). These lights proved to inactivate up to 95% of airborne viruses[71,72]. For the sake of simplicity, in simulations we reduce $p_\beta$ in 95% per contact, even though the relation between the local viral load and the contagion probability can be more intricate.

Regardless of the specific technology used for spatial immunization, the broader question addressed in this work is whether there exists an optimal spatial arrangement to minimize contagions given a certain number of cells to be immunized. This naturally deals with measuring the impact of each spatial immunization configuration in terms of the number of contagions in the airport, outbreak probability and destinations affected.

The areas to immunize are selected by the number of contagions predicted by the epidemic models. For each model configuration, we run 1000 realizations and rank the cells by the number of contagions occurring in them during the first day (see Supplementary Figs. S7-S9). The top ranking cells go under the name of infection hotspots, and their number can be varied depending on the amount of space that one is capable of covering with spatial immunization methods. There are 34792 cells in the model, we test the immunization of 50, 100, 200, 400 and 800 cells, corresponding respectively to 0.1%, 0.3%, 0.6%, 1.1% and 2.3% of the available space.

**Reporting summary**
Further information on research design is available in the Nature Portfolio Reporting Summary linked to this article.

## Data availability
Researchers may request access to Cuebiq data by submitting proposals through Cuebiq's "Spectus Social Impact" program (https://spectus.ai/social-impact/). Projects are considered on a case-by-case basis with priority given to projects that generate positive social impact and explore novel use cases and methodological lines of inquiry. Projects are subject to review by Cuebiq's Privacy Council, and may require additional review by Researcher institutions. Heathrow airport boundaries are available at https://www.openstreetmap.org/way/185882029#map=14/51.4693/-0.4537 under the license Open Data Commons Open Database Licence (ODbL).

## Code availability
Python2 was used for the analysis and models performed in this work. Codes employed in this work are publicly accessible on the Figshare repository at the link https://figshare.com/s/fe0a276da2cbca6d599d with https://doi.org/10.6084/m9.figshare.19780192[73].

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

## Acknowledgements

M.M. acknowledges financial support of the Sorbonne Université Emergence project RISKFLOW and SPHINX (ANR-17-CE36-0008-05). R.G. acknowledges the support of the PNRR ICSC National Research Centre for High Performance Computing, Big Data and Quantum Computing (CN00000013), under the NRRP MUR program funded by the NextGenerationEU. P.C. and J.J.R. acknowledge funding from project FACE by the European Commission - NextGenerationEU (Regulation EU 2020/2094), through CSIC's Global Health Platform (PTI Salud Global), from MCIN/AEI/10.13039/501100011033/FEDER/EU under project APASOS (PID2021-122256NB-C22) and from MCIN/AEI/10.13039/501100011033 under the Maria de Maeztu Program for units of Excellence in R&D CEX2021-001164-M. J.J.R. also acknowledges support from the PTI Mobility 2030 of CSIC.

## Author contributions

M.M., R.G., P.C. and J.J.R. conceived and designed the study. F.P. collected and R.G. pre-processed the trajectory data. M.M. and R.G. analyzed the trajectory data. M.M. and J.J.R. developed and adapted the models. M.M. performed the simulations. M.M., R.G., P.C. and J.J.R. wrote the paper. All the authors read and approved the paper.

## Competing interests

The authors declare no competing interests.
