## [Peer Review File · Nature Communications]

Spatial immunization to abate disease spreading in transportation hubsREVIEWER COMMENTS

Reviewer #1 (Remarks to the Author):

The authors use GPS data to reconstruct trajectories of persons (passengers and workers) in Heathrow airport. They identify contacts between persons if both persons are at the same time close together.

The authors superimpose different epidemic models to simulate the spread of (particularly airborne) infections. They also investigate the effect of reduction of transmission probabilities by disinfection (UV light) in certain locations.

I find the idea of the article interestingly, and suited for nat comm. The paper is not technically, well written and communicates (mostly) clearly.

I have one major concern:

The authors use parameters obtained in other papers by fitting deterministic epidemic models to data. As we have here a stochastic individual-based model, there is a potential to obtain misleading results: In a deterministic model, we basically assume exponentially distributed waiting times. If we estimate the parameters, we average (population size needs to be large for deterministic model, so averaging is OK). In an individual-based model, we need to be more precise.

E.g., the infectious period often is rather gamma distributed (with an appropriate dispersion). Potentially, the probability for a major outbreak might be different if the infectious period is differently distributed, also if the expected length of the infectious period is the same. See, e.g. [1].

Summary: All in all, I like the paper, and I think it is worth to be published in nat comm if the authors are able to comment on the concern above, and several minor remarks below.

[1] J. Wallinga, M. Lipsitch, How generation intervals shape the relationship between growth rates and reproductive numbers. Proc. R. Soc. B 274, 599–604 (2007)

Minor remarks:

p.2, l 22 Please explain/spell out abbreviations as "a GDPR compliant framework".

Any hints for some selection bias in the data? That other persons who are not covered (e.g., small children) would have different trajectories and in that would lead to different results?

p. 4, l1-10

Reference [60] is a preprint, published in 2010 (12 years ago). Either use a version published in a refereed journal, or rather another paper that did undergo a quality check?

p. 4 and later

Please be explicit in your parameter assignment (might be in the SI): Due to the contact network, you have an average contact rate (contacts per time interval) β , and you adapt p_{β} such that

$$R_0 = p_{\beta} \beta / \mu$$

where R_0 and μ is known from publications, and β from your empirical data. True?

p. 4

SIR models: Do worker stay at home if I (symptomatic cases)?

p. 4 Ebola

Please explain this example. Sounds not plausible to me, as you rightly explain, that's not an airborne disease. Simply passing-by in a way that is normal at an airport will hardly transmit the disease. I don't think this example should be included unless you give a distinct and persuading rational.

p 4, Covid-19

here you explicitly mention symptomatic and asymptomatic cases. How do you take that into account for working persons? They will most likely stay at home if they are symptomatic.

p.5 | 16

You might want to write here p_{β} instead of β . If you really want to have β here, please explain (in this case I don't understand what you mean).

To better interpret Fig 3:

Please,

(a) give the total number of cells (do you disinfect almost all cells, or only a tiny fraction?),
(b) draw the distribution no of contacts per cell (such that we can know if disinfect 400 cells, say, will eliminate all highly contagious cells or only a part of them).

Might be in the SI material.

p. 6 | 40

This two effects -> These two effects

Last remark:

I miss some further sensitivity analysis. As far as I understand, you construct the contact network once, based on specific movement patterns/trajectories.

If the initial time points of the trajectories are changed randomly, we obtain a different contact pattern.

Now comes the question.

If we identify the high-risk-sites with the first contact pattern, and then disinfect these sites, but then simulate the infection with the second contact pattern, do we still have the same impact?

That is, do we have some over-fitting to the specific contact pattern in the model in the identification of high-risk-spots, or is that effect quite stable?

Reviewer #2 (Remarks to the Author):

The authors investigate the effects of spatial immunization such as cleaning and disinfection to mitigate epidemics in transportation hubs. In this work, they use real mobility data of passengers and workers in London Heathrow airport. With this data, they were able to build a temporal co-presence network by reconstructing the paths of each individual inside the airport, classifying them as passengers or workers depending on where they pass. Basic epidemic models are used with different parameters to simulate different diseases such as SARS, influenza, COVID-19, and Ebola. The work is interesting, especially for the high-quality mobility data they show. However, I believe the authors could present the results in a better way than in this version, and a revision can be necessary. Also, since the method could be applied for other situations, as the authors say, at least a code/package would be welcome to facilitate the implementation. There are no details of how the simulations were implemented, for example.

Some questions:

- How about the bias in the mobility data? Do this data really represent the mobility patterns of individuals in the airport?

- There is a focus on general "immunization" (I do not know if it is the more accurate word) procedures, and then assume that they reduce around 95% of the contacts. It would be interesting to see results for other values than that, it is a large reduction.

- Related to the last point: What about the use of masks? This would be enforced in all cells, for example, and a more realistic assumption would be to assume the combination of different levels of "immunization" procedures.

- The authors perform a kind of supersampling that I think is fine. It is described in the SI, but I

did not get the following:

"For sake of consistency when running simulations longer than one day, we also want to avoid workers at midnight to appear as different users before and after 00 : 00." What does that mean?

- Also in the SI, I suggest adding something like "with a procedure described next" after the phrase "Although, we still have to filter our trajectories one more time."
- Please add what the color ranges mean in all Figures of the SI. The labels are present in the main paper, but not in the SI.
- Figure S2 is missing a and b labels.
- Typo in the SI: "data more then 10/20/..." on page 2.
- What is "user type assortativity" in the caption of Fig. 1?
- Also in the caption of Fig. 1: "they interact recurrently with each other": Is this a correct description of a connected graph?
- Please explain how the simulations were performed. It can be hard to apply the methodology if the readers do not know how to simulate the epidemics in these temporal networks.
- It was not clear to me how are the temporal networks. Are the connections aggregated for a given time window? Or are the connections explicitly followed when performing the simulations?
- How accurate are the copresence networks to model an Ebola transmission?
- Is there a relationship between the number of necessary cells to be immunized and a percolation process? Also, what happens to this number if the reduction of β is not so high as 95%?
- Please choose between "COVID", "Covid-19", or "COVID-19". The same happens for "Ebola" and "EBOLA".
- I missed some classical network measures like the degree distribution, for example, and burstiness for the temporal networks.
- I did not find it "remarkable" that the SIR model is very suitable to identify the ranking of places, it is already known in the literature since it captures the essence of the infection dynamics.
- The authors assume that the workers, even if infected, continue to go to the airport. Is that correct? If so, it does not sound realistic, since they could, for example, use masks if they continue working. Anyway, these "immunization" procedures do not seem so realistic to me.
- Finally, the authors say that these methodologies can be applied to other situations, but there is no code available. I understand that the data cannot be shared, but at least an "agnostic" code is welcome.

Reviewer #3 (Remarks to the Author):

In this article, Mazzoli et al. propose a method of spatially immunizing Heathrow airport to reduce the risk of transmission. The article highlights the ability of location data collected within busy international airports to identify hotspots of transmission within the airport and the potential benefits of focused immunization of these locations through the use of methods such as surface cleaning, air filtering, or UV lights. The article presents a ranking of cells within the airport for the

focused immunization under different disease scenarios, primarily focused on airborne diseases.

My comments on the article are as follows:

The model presented reconstructs trajectories of passengers and workers within the airport by supersampling over a period of 6 months with data collected by Cuebiq. This method for workers makes sense - they are likely to be well represented within the data in that they are likely to show up on more than one day in the dataset. For passengers, I think we need to understand how much of a sample the Cuebiq data likely represents - that is, how many passengers do travel within the airport during that time period?

Do you think the data collected and the supersampling method can account for flights with very high volumes of passengers? It seems that aggregating over 6 months means the model would be spreading contacts out and making the mean higher while cutting the tail of the distribution. We know that superspreading events are facilitated by a large number of contacts that can be exposed to disease. Can you speak to how more high density cells might change your results?

How do you account for gaps in data collection? For example, Cuebiq data is available because users have apps downloaded to their phones which then collect data and send to Cuebiq. Are there regions of the world where Cuebiq is known to undersample? Could this affect your results by underweighting the density of contacts within certain regions of the airport consistently used for flights from those regions of the world? In that case, could your model using this data create a bias in terms of identifying cells in the airport to strategically immunize? How could this bias be mitigated?

The section 'Simplest model: SIR' where the details of the simulation scenario are specified needs clarification. In particular, what does 'clone of the standard one as discussed above' mean? On the first day an infected individual lands at the airport at 13:30 local time. What's unclear to me is whether a new infected individual seeds the airport again on the next 7 days. I think not, but the language here should be clearer about what a clone of the standard day means.

Given that Ebola is much more obvious when symptomatic and the disease is transmitted through contact with fluids, and though it does progress more slowly, this disease does not seem to fit with diseases that would be more likely to spread at an airport. Fluid transmission means more concentrated contacts - i.e. with those you are close with or cleaning staff in the case of an airport. Ebola's impact would mean most airports are more likely to enact stricter policies than UV lights. I'm not sure it makes sense to present this scenario in the work. Are there other diseases with a slower progression that would fit better to make your case?

In addition to this, wouldn't frequent cleaning by staff increase the presence of workers in the airport in those contagion hotspots? If we are using contact rates in the model, should your work include an increase in presence due to cleaning frequency in the immunized cells or would those staff be wearing high levels of PPE to not add to the susceptible pool of the cell? I'm not sure it's feasible to model zero risk to those workers and thus, modeling them as not part of the human population in the immunized cells. It's interesting because this means that cleaning to some extent presents a contradiction in the immunization strategy.

The results presented here make reference to the number of cells immunized and the number of infections as a result. I think for the audience to make the most of these results it would make sense to also state how many cells there are total in the airport for context. Does 400 cells mean 5% of the airport or 20%? It's unclear at the moment how much of the airport might need to be immunized under these different scenarios and that would have an impact on the feasibility for implementation at different airports or related transportation hubs around the world.

What if the immunization methods are not as effective as 95%? Cleaning methods might not be as effective due to the lack of quality of filtering or cleaning. In this case, have you considered a lower efficacy scenario? How might this change the timing of peak cases in your results?

In practice, how would an airport even know of an outbreak occurring within their walls so early on? The scenarios presented here look at the results of immunizing within the first few days of the outbreak. Have you considered scenarios where outbreak identification happens later on? How quickly does the airport need to identify the outbreak among, for example their staff who might have reasons to not report symptoms? What is the efficacy of these immunization strategies if not everyone reports when they are infected?

Beyond this, I think a check for grammar and spelling would benefit the article's readability, but overall I find the article is clearly laid out and interesting to read, even if I'm not sure that it would be a feasible approach for airports in practice.

REVIEWER COMMENTS

Reviewer #1 (Remarks to the Author):

The authors use GPS data to reconstruct trajectories of persons (passengers and workers) in Heathrow airport. They identify contacts between persons if both persons are at the same time close together.

The authors superimpose different epidemic models to simulate the spread of (particularly airborne) infections. They also investigate the effect of reduction of transmission probabilities by disinfection (UV light) in certain locations.

I find the idea of the article interestingly, and suited for nat comm. The paper is not technically, well written and communicates (mostly) clearly.

I have one major concern:

The authors use parameters obtained in other papers by fitting deterministic epidemic models to data. As we have here a stochastic individual-based model, there is a potential to obtain misleading results: In a deterministic model, we basically assume exponentially distributed waiting times. If we estimate the parameters, we average (population size needs to be large for deterministic model, so averaging is OK). In an individual-based model, we need to be more precise.

E.g., the infectious period often is rather gamma distributed (with an appropriate dispersion). Potentially, the probability for a major outbreak might be different if the infectious period is differently distributed, also if the expected length of the infectious period is the same. See, e.g. [1].

We thank the reviewer for this valuable comment to our methodology. While population models rely on exponential waiting times for infectious periods, here we deal with infections occurring at individual level. Following the reviewer's suggestion, we have performed new simulations using Gamma-distributed infectious periods in order to correct for this inaccuracy and we have replaced all the respective results in the main text and the SI. As can be seen, the main results remain qualitatively invariant after the model update, which further contribute to our certainty in the robustness of these results.

Summary: All in all, I like the paper, and I think it is worth to be published in nat comm if the authors are able to comment on the concern above, and several minor remarks below.

[1] J. Wallinga, M. Lipsitch, How generation intervals shape the relationship between growth rates and reproductive numbers. Proc. R. Soc. B 274, 599–604 (2007)

We thank the reviewer for the positive opinion and the valuable contribution, we hope that the new version of our manuscript will now meet the requirements for acceptance.

Minor remarks:

p.2, l 22 Please explain/spell out abbreviations as "a GDPR compliant framework".

We added the complete name of this acronym, *General Data Protection Regulation*, in the text.

Any hints for some selection bias in the data? That other persons who are not covered (e.g., small children) would have different trajectories and in that would lead to different results?

In order to answer this question, we have to look into the data from the Civil Aviation Authority (https://www.caa.co.uk/media/zs0lcx3t/t11_2018.pdf , Table 11.7 for Heathrow airport). These data concern the composition by age of passengers in Heathrow airport. We can see that people aged less than 19 years old represent approximately only 5% of total passengers, while people aged 65+ represent approximately 10% of the total passengers. In general, most passengers are included in the range 25-54 years old. By looking at the percentage of smartphones usage in the UK stratified by age (<https://cybercrew.uk/blog/smartphone-usage-statistics-uk/>), we can see that the usage is pretty uniform in the spectrum ranging from 25-54 years old. From a mobility perspective, young children will always be accompanied by adults, i.e. their path will never be a completely different one with respect to the one of their parents/accompanists. From an epidemiological perspective, this results in strongly coupled trajectories between children and accompanists, hence strongly coupled disease statuses. By construction of our supersampling process, coupled trajectories are already present in our network. For all these reasons, we are confident that age bias is not a major concern for this type of study in these particular settings, i.e. transportation hubs. However, this may be an issue when looking at schools or hospitals, where population composition is much more represented by children and elders.

For details on coverage bias in other parts of the world and how to mitigate it, see

<https://spectus.ai/social-impact/lets-talk-about-bias/>

We added a subsection called *Data representativeness* in the SI discussing this topic.

p. 4, 11-10

Reference [60] is a preprint, published in 2010 (12 years ago). Either use a version published in a refereed journal, or rather another paper that did undergo a quality check?

We thank the reviewer for highlighting this. We removed Ref[60], we now refer to two published works for modeling SARS

[61]- "Zhang, Zhibin. "The outbreak pattern of SARS cases in China as revealed by a mathematical model." *Ecological Modelling* 204.3-4 (2007): 420-426."

[64]- "Riley S, Fraser C, Donnelly CA, Ghani AC, Abu-Raddad LJ, Hedley AJ, Leung GM, Ho LM, Lam TH, Thach TQ, Chau P. Transmission dynamics of the etiological agent of SARS in Hong Kong: impact of public health interventions. *Science*. 2003 Jun 20;300(5627):1961-6" and re-run all simulations with the new parameters.

p. 4 and later

Please be explicit in your parameter assignment (might be in the SI): Due to the contact network, you have an average contact rate (contacts per time interval) β , and you adapt p_β such that

$$R_0 = p_\beta \beta / \mu$$

where R_0 and μ is known from publications, and β from your empirical data. True?

The optimal way to parametrize the model would be to have the estimation of R_0 and the contact rate from the literature for the population used in those studies. In this way, we could estimate p_β per contact and inform the model with a contagion probability consistent with the original studies. R_0 is usually estimated with statistical models at population level, not considering individual level heterogeneities such as different individual contact rates. From a technical point of view, the contact rate observed in the airport for each individual is not representative of their contact rate in other settings. Using the contact rate measured in the airport without considering that this applies only for a fraction of their total infectious period would lead to underestimate the probability of infection p_β of the airport population.

Given that we need to use a certain value of p_β , we take two approaches: The first is to estimate p_β in order to recover the values of R_0 reported in the literature inside the airport facilities. This is done by informing the model with the contact rate β empirically measured in the airport to estimate the average transmissibility (or probability of infection per contact) p_β . Given an estimation of R_0 for a certain disease, we take into account two factors for the parameterization of p_β in the model:

- Airports are open systems, the population is never conserved in time, people arrive and leave, changing the total susceptible population inside the system at all hours and, most importantly, reducing the real time of infectiousness of agents in the system. For a given R_0 , infectious agents would be able to infect susceptibles only when they are present in the airport, reducing the number of secondary infections due to anticipated removal. To correct for this, we need to take into account the median fraction of time spent in the airport by all individuals (new Fig.S4a), $f = 8 \text{ time slots} / 96 \text{ time slots} = 2 \text{ hours} / 24 \text{ hours} = 0.083$

Fig.S4a: Time spent by agents in the airport perimeter during the standard day expressed in 15' time slots. Whiskers represent the 5-95% interval, boxes the IQR, red line the median. Most agents spend less than 7 hours within the perimeter, but the distribution is highly skewed due to the workers, connecting and delayed passengers.

- Airports are very crowded settings in which individuals produce many more contacts than the average general population and, hence, are not representative of an average individual's behavior. Moreover, these contacts are produced only in a fraction of the whole day, and hence, only during a fraction of the individual's infectious period. The number of unique contacts made in a time slot by each individual is highly heterogeneous along the day and per individual, hence we compute the average of the median contact rates at all time slots (blue solid line), $\beta = 34.6 \text{ contacts/time slot}$, as shown in new Fig. S4b. Since we do not have information on the contacts produced outside of the airport by the same individuals, we can correct this rate by the above measured fraction f of daytime spent in the airport. This leads to the adjusted contact rate $\beta' = 2.89 \text{ contacts/time slot}$.

Fig.S4b: Contact rate of agents along the standard day. Blue line represents the median of the distribution obtained from all agents' number of unique contacts at each time slot. Shaded area stands for the 5-95% interval of the distribution obtained from all agents' number of unique contacts at each time slot.

Finally $R_0 = p_\beta \beta' / \mu$, hence $p_\beta = R_0 \mu / \beta'$ where μ is the probability of recovery expressed in 15-minutes time slots rather than in days, as typically reported in the literature.

We know that this is a very rough approximation to the real p_β of a given disease. Therefore, the second approach consists in a sensitivity analysis to make sure that our identification of the contagion hotspots and transmission reductions are robust. We performed further simulations for SARS-CoV2 exploring a range of different transmissibility values, with lower and higher values for p_β , recovering qualitatively similar results. In all the cases, the spatial immunization policy produced a sensible reduction of outbreak intensity. We added a clarification about this important point in the SI. in Secs. *Modeling epidemics in open systems and Generalizability of the method*, and also in the discussion of the main text.

p. 4

SIR models: Do worker stay at home if I (symptomatic cases)?

In the SIR model, we do not distinguish between symptomatic and asymptomatic infected and workers are not required to stay at home if infected. Note that here we designed a scenario of a first undetected outbreak in the absence of any other intervention. Agents and public health authorities in this scenario are assumed to be unaware that cases of a new disease are arriving at the airport, hence business run as usual and no test-isolation protocols are implemented. This would eventually occur, but a little later. Since we have complete knowledge of all contagions in the model, we are able to assess what would be

the advantage of having immunization policies (the ones we test or others) implemented in the airport. In brief our idea is to count with a background silent protocol of outbreak protection active at all times. The importance of this relies on its ability to reduce contagion probability and outbreak intensity in the case of lack of detection of the first imported cases. We added this clarification in the last paragraph of the introduction.

p. 4 Ebola

Please explain this example. Sounds not plausible to me, as you rightly explain, that's not an airborne disease. Simply passing-by in a way that is normal at an airport will hardly transmit the disease. I don't think this example should be included unless you give a distinct and persuading rational.

The reviewer is right that a major difference exists between Ebola and other infectious diseases that we explored in our study. Ebola was included in the first place since at-risk contacts in the airport occur not only via aerosol but also through materials and surfaces carrying body fluids like sweat or droplets. Our co-presence definition was thought as an approximation that could include these other types of contact, but we understand that using the same framework and parameterization for Ebola as for SARS and H1N1 is too far-fetched, hence we removed this case study from our manuscript.

p 4, Covid-19

here you explicitly mention symptomatic and asymptomatic cases. How do you take that into account for working persons? They will most likely stay at home if they are symptomatic.

Similarly as for the SIR case, we assume that agents and public health authorities are completely unaware of the developing disease outbreak in the airport, hence no test-isolation protocols are implemented in the model. We designed the study thinking in a scenario in the first days of an undetected disease outbreak in order to model what would be the effect of a spatial immunization in the most critical stage of a new starting epidemic in absence of any other intervention. Of course, in the case of a recognized disease outbreak, public health authorities would issue protocols requiring workers and all individuals to stay at home if testing positive. These protocols could be implemented in our model and it could be an interesting continuation to the present manuscript, but here we prefer to focus on the very early stages of an emerging disease propagation. As a real-world example, this would equal to study the spreading of Covid-19 in December 2019-January 2020. Individuals would feel sick but would be unaware of the kind of disease that they are experiencing.

p.5 1 16

You might want to write here β_{p} instead of β . If you really want to have β here, please explain (in this case I don't understand what you mean).

We thank the reviewer for noticing this error, it is indeed p_{beta} that is reduced. We corrected this in the text.

To better interpret Fig 3:

Please,

(a) give the total number of cells (do you disinfect almost all cells, or only a tiny fraction?),

There are 34792 cells available in the system as a result of filtering out those cells that did not register more than 30 presences. Hence, every time we immunize 50, 100, 200, 400, 800 cells, we are actually immunizing 0.1%, 0.3%, 0.6%, 1.1%, 2.3% of the available space in the model. We added this clarification in the last paragraph of the Methods section in the main text.

(b) draw the distribution of contacts per cell (such that we can know if disinfect 400 cells, say, will eliminate all highly contagious cells or only a part of them).

Might be in the SI material.

Thanks for the suggestion, we added to the SI the new Fig S5 here below.

Fig. S5: Number of unique contacts occurring per cell. a) Distribution of the number of unique contacts occurring in each cell during the day. Dashed lines represent respectively the top 50, 100, 200, 400, 800 ranked cells by number of contacts in one day. b) Distribution of the number of unique contacts occurring in each cell at each time slot. Dashed area represents the 5-95% interval, solid line represents the median.

What can be observed is that the distribution of total contacts occurring in each cell is highly heterogeneous. The top ranked cells by number of contacts do not exactly correspond to the top ranked cells by number of contagions, since also the duration of the contacts plays an important role. From this distribution, we see that most at-risk contacts

concentrate in a few cells of the airport. This explains the high efficacy of our spatial immunization method even with percentages of immunized cells lower than 2% (<800 cells).

p. 6 | 40

This two effects -> These two effects

We corrected this in the text.

Last remark:

I miss some further sensitivity analysis. As far as I understand, you construct the contact network once, based on specific movement patterns/trajectories.

If the initial time points of the trajectories are changed randomly, we obtain a different contact pattern.

Now comes the question.

If we identify the high-risk-sites with the first contact pattern, and then disinfect these sites, but then simulate the infection with the second contact pattern, do we still have the same impact?

That is, do we have some over-fitting to the specific contact pattern in the model in the identification of high-risk-spots, or is that effect quite stable?

We thank the reviewer for remarking this potential issue. In order to check for this, we applied a user-wise normal noise to trajectories in the dataset with mean $\mu = 0'$ and standard deviation $\sigma = 15'$ and created a second contact network. We ran our models in the new contact network and found consistent results with the original network, see new Fig S14. Note that mobility in the airport can be slightly altered, however users' trajectories are strongly constrained by flight schedule, airport closure hours and individuals' personal purpose within the airport. Hence altering of users' trajectories must be applied with caution in order to preserve realistic traffic flows within the airport corridors. We have added these tests of robustness in the new version of the SI.

Reviewer #2 (Remarks to the Author):

The authors investigate the effects of spatial immunization such as cleaning and disinfection to mitigate epidemics in transportation hubs. In this work, they use real mobility data of passengers and workers in London Heathrow airport. With this data, they were able to build a temporal co-presence network by reconstructing the paths of each individual inside the airport, classifying them as passengers or workers depending on where they pass. Basic epidemic models are used with different parameters to simulate different diseases such as SARS, influenza, COVID-19, and Ebola. The work is interesting, especially for the high-quality mobility data they show. However, I believe the authors could present the results in a better way than in this version, and a revision

can be necessary. Also, since the method could be applied for other situations, as the authors say, at least a code/package would be welcome to facilitate the implementation. There are no details of how the simulations were implemented, for example.

We thank the reviewer for the positive opinion and the valuable contribution, we hope that the new version of our manuscript will meet the requirements for acceptance. We have now included a commented open code describing all the steps in the model used in the manuscript (<https://figshare.com/s/fe0a276da2cbca6d599d>).

Some questions:

- How about the bias in the mobility data? Do this data really represent the mobility patterns of individuals in the airport?

In order to answer this question, we have to look into the data from the Civil Aviation Authority (https://www.caa.co.uk/media/zs0lcx3t/t11_2018.pdf, Table 11.7 for Heathrow airport). These data concern the composition by age of passengers in Heathrow airport. We can see that people aged less than 19 years old represent approximately only the 5% of total passengers, while people aged 65+ represent approximately the 10% of total passengers. In general, most passengers are included in the range 25-54 years old. By looking at the percentage of smartphones usage in UK stratified by age (<https://cybercrew.uk/blog/smartphone-usage-statistics-uk/>), we can see that the usage is pretty uniform in the spectrum ranging from 25-54 years old. From a mobility perspective, young children will always be accompanied by adults, i.e. their path will never be a completely different one with respect to the one of their parents/accompanists. From an epidemiological perspective, this results in strongly coupled trajectories between children and accompanists, hence strongly coupled epidemiological statuses. By construction of our supersampling process, coupled trajectories are already present in our network. For all these reasons, we are confident that age bias is no concern for this type of study in this particular setting, i.e. transportation hubs. However, this may be an issue when looking at schools or hospitals, where age composition is much more represented by children and elders. We added a subsection called *Data representativeness* in the SI discussing this topic.

- There is a focus on general "immunization" (I do not know if it is the more accurate word) procedures, and then assume that they reduce around 95% of the contacts. It would be interesting to see results for other values than that, it is a large reduction.

We performed further simulations assuming a lower reduction on transmissibility due to the process of immunization, respectively for 80% and 65%. We added these results in the SI, see new Figs.S12-S13. In recent experiments applying UV-C lamps, researchers showed that a reduction of 93-98% of the pathogen load of airborne pathogens can be achieved in the context of a room. See Eadie, Ewan, et al. "Far-UVC (222 nm) efficiently

inactivates an airborne pathogen in a room-sized chamber." *Scientific Reports* 12 (2022): 4373 (<https://www.nature.com/articles/s41598-022-08462-z#Sec2>). This new empirical evidence has been added to the present version of the manuscript.

- Related to the last point: What about the use of masks? This would be enforced in all cells, for example, and a more realistic assumption would be to assume the combination of different levels of "immunization" procedures.

Here we assume that agents and public health authorities are completely unaware of the developing disease outbreak in the airport. We designed the study thinking of a what-if scenario as the first days of an undetected disease outbreak in order to model what would be the effect of a spatial immunization in absence of any other intervention. Of course, in the case of a recognized disease outbreak, public health authorities would develop protocols requiring prophylactic measures as mask use. However, this is not the stage of the epidemic that we want to study here. Such scenarios could be analyzed with a further development of the model, but considering them here would hinder the understanding of the effectiveness of preventive spatial immunization. This later stage analysis would certainly be something that would like to be explored in a follow-up work. We added a clarification regarding the context of our scenario in the last paragraph of the introduction.

- The authors perform a kind of supersampling that I think is fine. It is described in the SI, but I did not get the following:
"For sake of consistency when running simulations longer than one day, we also want to avoid workers at midnight to appear as different users before and after 00 : 00." What does that mean?

We thank the reviewer for noticing that this sentence was unclear. Night shift workers appear both in the morning and night due to their shift starting before midnight and ending after midnight. Since they appear regularly day after day, their trajectory does not require further treatment in order to be counted in the standard day. We removed this sentence from the text since it was ambiguous and redundant.

- Also in the SI, I suggest adding something like "with a procedure described next" after the phrase "Although, we still have to filter our trajectories one more time."

We added this sentence in the text.

- Please add what the color ranges mean in all Figures of the SI. The labels are present in the main paper, but not in the SI.

We thank the reviewer for noticing this missing information. We added the respective descriptions of color code in all the missing figures of the SI.

- Figure S2 is missing a and b labels.

We corrected this.

- Typo in the SI: "data more then 10/20/..." on page 2.

Thank you for noticing this typo, we corrected it.

- What is "user type assortativity" in the caption of Fig. 1?

This is the Attribute Assortativity Coefficient, it measures how much nodes of the same type are linked to each other with respect to the overall types of nodes in the network. In our case it measures how much users of the same category mix with each other from the network aggregated over one month of data. Assortativity measures the correlation of contacts patterns in terms of nodes degree. If a network is assortative, nodes with high degree will be more linked with nodes of high degree, and vice versa. Similarly, here the interpretation of this score is that users being workers will be much more in contact with workers, whereas passengers will be more in contact with passengers. See Ref <https://networkx.org/nx-guides/content/algorithms/assortativity/correlation.html>

- Also in the caption of Fig. 1: "they interact recurrently with each other": Is this a correct description of a connected graph?

The reviewer is right that this is rather a behavioral explanation that has repercussions on the topology of the network and not vice versa. Here we aggregated the temporal network over one month of data. Hence, we have a static network and we can measure how much users of the same category are linked to each other with respect to other categories. An explanation of why workers are strongly linked to each other is that they attend workspaces more often in company of their colleagues and spend much more time in the airport with respect to passengers. Their interactions are thus recurrent, giving rise to more ties between workers in the aggregated network with respect to other categories.

- Please explain how the simulations were performed. It can be hard to apply the methodology if the readers do not know how to simulate the epidemics in these temporal networks.

Further explanations on the typical simulation steps have been added in the SI section *Modeling epidemics in open systems*. Beside, an agnostic code of the simulations with all the necessary step by step explanations has been uploaded at the link <https://figshare.com/s/fe0a276da2cbca6d599d>

- It was not clear to me how are the temporal networks. Are the connections aggregated for a given time window? Or are the connections explicitly followed when performing the simulations?

The connections in the temporal network of the standard day are aggregated for a given time window, as correctly stated by the reviewer. This basic time window is of 15 minutes and it represents the maximum granularity achieved from the data in order to define co-presences in the airport. All co-presences times are thus assumed to be integer multiples of the basic 15-minutes slot. The temporal network of the standard day is defined once and for all, and repeated day by day to reach one week of modeling. When running the models, the order of connections in the standard day is preserved in all simulations.

- How accurate are the copresence networks to model an Ebola transmission?

As observed by another reviewer, it is true that a major difference exists between Ebola and other infectious diseases that we explored in our study. Ebola was included in the first place since at-risk contacts in the airport occur not only via aerosols (airborne), but also through materials and surfaces, hence carrying body fluids like sweat or droplets. Our co-presence definition can be also used as a rough approximation for these types of contacts, and the intention was to offer an example with a disease with an alternative form of contact. However, we understand that using the same framework and parameterization for Ebola as for SARS and H1N1 can lead to confusion and we have removed this case study from our manuscript.

- Is there a relationship between the number of necessary cells to be immunized and a percolation process? Also, what happens to this number if the reduction of β is not so high as 95%?

This is a very interesting question, and it is something that we have thought about as well. However, the research question is not quite the same.

- Note that here we do not try to compute the necessary number of cells to be immunized in order to suppress epidemic spreading, since epidemic spreading cannot be completely suppressed. Even immunizing the whole airport, an imported case always has a finite probability of leading to a secondary infection. The more the number of immunized cells, the less probable it is. Identifying the number and the cells to be immunized is an equation in which one must consider resource allocation and costs to be assumed by the airport management vs potential benefits.

- Although percolation concepts could be used, the cells to be immunized are usually correlated in space, it is not a simple case of uncorrelated spatial percolation. Assuming a spatial network in which nodes are cells and links are weighted by the number of consecutive visits by the same individuals, by performing random extractions from the set of cells in the airport, we would gradually recover the original spatial network by tuning the probability parameter ϕ . When reaching a certain threshold ϕ_c , a giant component would appear. Technically, spreading would occur even before reaching the giant component, but contagions would be highly constrained to specific areas, not representing realistic transmission routes. Finally, the threshold ϕ_c for obtaining the giant component spread in the network will not be connected to the fraction of cells defined as hotspots, since hotspots are located in the fat-tail of the distribution of contacts per cell (see new Fig.S5a) and are not identified randomly in space. Moreover, there is spatial correlation among cells in terms of number of contacts, e.g. hotspots neighboring cells will easily exhibit high numbers of contacts. Finally, the fraction of nodes to randomly remove in order to recover our modeled disease spreading disruption would be much higher than the one that we defined. This is due to the distribution of contacts per cell in Fig.S5a. As a result, a strategy like ours, based on the so-called targeted attacks on the spatial network, results in much more effective disruption of transmission routes in the network.

Fig. S5a: Number of unique contacts occurring per cell. Distribution of the number of unique contacts occurring in each cell during the day. Dashed lines represent respectively the top 50, 100, 200, 400, 800 ranked cells by number of contacts in one day.

In order to answer the second question, we performed further simulations assuming a lower reduction on transmissibility due to the process of immunization, respectively for 80% and 65%. We added these results in the SI, see new Figs.S12-S13. Note that in recent experiments applying UV-C lamps, researchers showed that a reduction of 93-98% of the pathogen load of airborne pathogens can be achieved in the context of a room. See

Eadie, Ewan, et al. "Far-UVC (222 nm) efficiently inactivates an airborne pathogen in a room-sized chamber." *Scientific reports* 12.1 (2022): 4373. (<https://www.nature.com/articles/s41598-022-08462-z#Sec2>). However, lower efficacies of this method could be achieved if implementing different techniques rather than UV-C lamps.

- Please choose between "COVID", "Covid-19", or "COVID-19". The same happens for "Ebola" and "EBOLA".

We thank the reviewer for noticing these inconsistencies. We fixed them all in the text.

- I missed some classical network measures like the degree distribution, for example, and burstiness for the temporal networks.

We added measures of degree distribution, number of unique contacts occurring in each cell per day and burstiness (as interval between active times of a same link) in the SI, see new figures Figs. S4-S5

Figures S4-S5: Contact heterogeneities. a) Contact rate of agents along the standard day. Blue line represents the median of the distribution obtained from all agents' number unique contacts at each time slot, shaded area stands for the 5-95% interval of the distribution obtained from all agents' number of unique contacts at each time slot. a) Distribution of the number of unique contacts occurring in each cell during the day. Dashed lines represent respectively the top 50, 100, 200, 400, 800 ranked cells by number of contacts in one day. c) Distribution of the interval between activation times of a same link along the standard day expressed in time slots of 15 minutes.

- I did not find it "remarkable" that the SIR model is very suitable to identify the ranking of places, it is already known in the literature since it captures the essence of the infection dynamics.

We agree with the reviewer that it is not remarkable that the SIR is the most suitable model to capture the essence of transmission dynamics. Our results confirm this knowledge. The important result here is that even an easy model such as the SIR, which does not always mimic disease progression, can be used to suggest targeted interventions in multiple contexts for implementing policies in a realistic scenario. We added this point in the discussion.

- The authors assume that the workers, even if infected, continue to go to the airport. Is that correct? If so, it does not sound realistic, since they could, for example, use masks if they continue working. Anyway, these "immunization" procedures do not seem so realistic to me.

Here we assume that agents and public health authorities are completely unaware of the developing disease outbreak in the airport. We designed the study as a what-if scenario of the first days of an undetected disease outbreak. We did so in order to model what would be the effect of a spatial immunization in absence of any other intervention. Of course, in the case of a recognized disease outbreak, public health authorities would develop protocols requiring mask mandates or other prophylactic measures. However, this is not the stage of the epidemic that we want to test here. Including more levels of intervention here would make the model more complex and would hinder its interpretability and the understanding of the effectiveness of the spatial immunization that we are testing. However, this is certainly something that could be tested in a follow-up work.

- Finally, the authors say that these methodologies can be applied to other situations, but there is no code available. I understand that the data cannot be shared, but at least an "agnostic" code is welcome.

We have uploaded the code for the epidemic model and the supersampling process, it is available at this link <https://figshare.com/s/fe0a276da2cbca6d599d>

Reviewer #3 (Remarks to the Author):

In this article, Mazzoli et al. propose a method of spatially immunizing Heathrow airport to reduce the risk of transmission. The article highlights the ability of location data collected within busy international airports to identify hotspots of transmission within the airport and the potential benefits of focused immunization of these locations through the use of methods such as surface cleaning, air filtering, or UV lights. The article presents a ranking of cells within the airport for the focused immunization under different disease scenarios, primarily focused on airborne diseases.

My comments on the article are as follows:

The model presented reconstructs trajectories of passengers and workers within the

airport by supersampling over a period of 6 months with data collected by Cuebiq. This method for workers makes sense - they are likely to be well represented within the data in that they are likely to show up on more than one day in the dataset. For passengers, I think we need to understand how much of a sample the Cuebiq data likely represents - that is, how many passengers do travel within the airport during that time period?

In order to answer this question at least in the age-spectrum representativity, we have to look into the data from the Civil Aviation Authority (https://www.caa.co.uk/media/zs0lcx3t/t11_2018.pdf , Table 11.7 for Heathrow airport). These data concern the composition by age of passengers in Heathrow airport. We can see that people aged less than 19 years old represent approximately only 5% of total passengers, while people aged 65+ represent approximately 10% of total passengers. In general, most passengers are included in the range 25-54 years old. By looking at the percentage of smartphones usage in UK stratified by age (<https://cybercrew.uk/blog/smartphone-usage-statistics-uk/>), we can see that the usage is pretty uniform in the spectrum ranging from 25-54 years old and well above 80%. Since passengers age distribution is clearly peaked within the working age range, smartphones GPS traces shall be captured well both for passengers and workers. Hence, we have no reason to think that relevant categorical bias between workers and passengers in the airport may affect our results. We added a subsection called *Data representativeness* in the SI discussing this topic.

Another aspect of this question is whether the trajectories of the cumulated day are representative of a single day. It is true that workers are easier to detect. Still, we have applied the supersampling method separately to both populations in order to reproduce the daily numbers observed in Heathrow in the official statistics. For passengers, flight operations have a strong (in many cases daily) periodicity and this gives us some confidence on the representativity of passenger trajectories. Even if we do not capture all the passengers, we observe trajectories of people performing similar trajectories, as they are strongly constrained by airport control areas, terminal structure and gates position. Hence we are confident of their representativeness of the common behavior. This is an assumption that is explained in the paper and we do not have a clear alternative to it, more powerful datasets with all the individuals detected would avoid this cumulative process.

Do you think the data collected and the supersampling method can account for flights with very high volumes of passengers? It seems that aggregating over 6 months means the model would be spreading contacts out and making the mean higher while cutting the tail of the distribution. We know that superspreading events are facilitated by a large number of contacts that can be exposed to disease. Can you speak to how more high density cells might change your results?

The reviewer is right when saying that superspreading events occur more often within high density cells. Hotspots are, with some degree of approximation in the definition of the term, the location where *superspreading* events occur most probably.

- Regarding the first point on high volumes of passengers flights: note that we do not go too much in detail as to look at what happens within aircrafts, the data granularity does not allow for it, hence in a majority of cases, the contagions in our model are occurring within the airport corridors, warehouses, workplaces, control areas, restauration areas and waiting rooms. Speaking about flight passenger volumes, we have no reason to think that the data may be biased towards low passenger volume flights. A percentage of the airport users all along the 6 months is captured depending on whether they use specific apps in their smartphones or not. We have no reason to think that users of smartphone apps may preferentially travel in big aircrafts rather than in small ones. For this reason, we are confident that no significant bias regarding flights with low or high passengers volumes may affect our results.
- From a more general perspective, high density cells play a crucial role in our models, this is represented in new Fig. S8, in which we can see that many of the SIR-designed hotspots overlap with hotspots designed solely on the number of presences in the standard day. Still, this overlap does not reach 100% since SIR-designed hotspots take into account recurrency and duration of contacts between workers and towards passengers, which a pure presence analysis could not capture. Having more heterogeneity in the distribution of cells density of contacts would ease the identification of very important hubs playing a major role in superspreading events, increasing the efficiency of our method for small percentages of immunized space, hence reducing the need for further immunization of higher percentages of the available space.

These points have been further discussed in the Sec. Contact patterns heterogeneities of SI and mentioned the main text.

How do you account for gaps in data collection? For example, Cuebiq data is available because users have apps downloaded to their phones which then collect data and send to Cuebiq. Are there regions of the world where Cuebiq is known to undersample? Could this affect your results by underweighting the density of contacts within certain regions of the airport consistently used for flights from those regions of the world? In that case, could your model using this data create a bias in terms of identifying cells in the airport to strategically immunize? How could this bias be mitigated?

The reviewer is right that Cuebiq data may be underrepresenting passengers from other parts of the world. This may bring a slight bias to our analysis. However, we inferred destinations of infected agents by looking at the probability of flying from Heathrow

terminals to countries of destinations from air traffic data. Air traffic records rely on official schedules of Heathrow airports hence shall not be affected by this kind of bias. Passengers flying to Cuebiq underrepresented regions of the world may be both residents of those regions or UK residents, and these latter users are not missing from the database. However, by looking at terminals' flights composition, we could not spot strong associations between airport terminals locations and traffic regarding specific world regions. Gates are not specifically dedicated to one unique region of the world. Hence, arrival passengers' nationalities will be generally mixed in all areas of the airport, therefore underweighting of contacts shall not affect specific areas of Heathrow airport. On the other hand, we have no information on incoming passengers regarding their flight origin from the mobility data, hence we cannot infer passenger's origins from their first appearance to correct for this potential misrepresentation. As a last remark, note that here we aggregated passengers' destinations into UK (domestic), European Union, London (incoming) and Intercontinental travelers. The latter includes the Americas, Asia, Africa and Oceania. Hence underrepresented areas of the world may once more be mixed to well-represented areas of the world. For this reason, we are confident that this kind of potential bias in Cuebiq data shall not affect our results too much. For details on coverage bias in other regions of the world and how to mitigate it, see <https://spectus.ai/social-impact/lets-talk-about-bias/>

A discussion on this potential bias has been included in the Sec. Data representativeness of the SI.

The section 'Simplest model: SIR' where the details of the simulation scenario are specified needs clarification. In particular, what does 'clone of the standard one as discussed above' mean? On the first day an infected individual lands at the airport at 13:30 local time. What's unclear to me is whether a new infected individual seeds the airport again on the next 7 days. I think not, but the language here should be clearer about what a clone of the standard day means.

The reviewer is correct saying that this sentence was ambiguous. The seed appears only and exclusively on the first day. Only the sequence of contacts between agents in the temporal network is repeated day after day. We corrected this in the manuscript.

Given that Ebola is much more obvious when symptomatic and the disease is transmitted through contact with fluids, and though it does progress more slowly, this disease does not seem to fit with diseases that would be more likely to spread at an airport. Fluid transmission means more concentrated contacts - i.e. with those you are close with or cleaning staff in the case of an airport. Ebola's impact would mean most airports are more likely to enact stricter policies than UV lights. I'm not sure it makes sense to present this scenario in the work. Are there other diseases with a slower progression that would fit better to make your case?

As observed by the other reviewers, it is true that a major difference exists between Ebola and other infectious diseases that we explored in our study. Ebola was included in the first place since at-risk contacts in the airport occur not only via aerosol but also through materials and surfaces, hence carrying body fluids like sweat or droplets. Our co-presence contact definition can handle in a rough approximation both airborne and surface-based contacts, but we understand that using the same framework and parameterization for Ebola as for SARS and H1N1 can lead to confusion, and we have, therefore, removed this case study from our manuscript. Other diseases with slow progression may recently include Monkeypox, however, even in this case human transmission occurs mostly via body contacts and fluids, so once again it is not strictly appropriate to use the same contact definition and parameterization of models as for the case of respiratory infectious diseases.

In addition to this, wouldn't frequent cleaning by staff increase the presence of workers in the airport in those contagion hotspots? If we are using contact rates in the model, should your work include an increase in presence due to cleaning frequency in the immunized cells or would those staff be wearing high levels of PPE to not add to the susceptible pool of the cell? I'm not sure it's feasible to model zero risk to those workers and thus, modeling them as not part of the human population in the immunized cells. It's interesting because this means that cleaning to some extent presents a contradiction in the immunization strategy.

This is an interesting and non-trivial point that would deserve to be explored in a dedicated work. It is true that including extra cleaning staff intrinsically introduces further susceptible populations into the system that may play a role in the spread of diseases. In our models, we are assuming a reduction of transmissibility in hotspots that may be achieved by different techniques. Intensive cleaning, rather than mask mandates or UV lights, is only one of the possibilities. In the case of interpreting this reduction as due to intensive cleaning, it is fair to assume cleaning staff as wearing high levels PPE, since in our models we are not adding further agents to the susceptible pool. However, what the reviewer mentions must be taken into account at the moment of issuing this specific technique, since introducing extra staff in hotspots areas may be counterproductive in terms of outbreak probability reduction. One solution could be to have this type of targeted cleaning carried out by already hired staff, who already carry out similar mansions in the airport at similar hours, possibly by wearing professional PPE.

The results presented here make reference to the number of cells immunized and the number of infections as a result. I think for the audience to make the most of these results it would make sense to also state how many cells there are total in the airport for context. Does 400 cells mean 5% of the airport or 20%? It's unclear at the moment how much of the airport might need to be immunized under these different scenarios and that would have an impact on the feasibility for implementation at different airports or related

transportation hubs around the world.

We thank the reviewer for noticing this lack of clarity. There are 34792 cells available in the system as a result of filtering out those that did not register more than 30 presences. Hence every time we immunize 50, 100, 200, 400, 800 cells, we are actually immunizing 0.1%, 0.3%, 0.6%, 1.1%, 2.3% of the available space in the model. We added this clarification in the last paragraph of the Methods section.

What if the immunization methods are not as effective as 95%? Cleaning methods might not be as effective due to the lack of quality of filtering or cleaning. In this case, have you considered a lower efficacy scenario? How might this change the timing of peak cases in your results?

We performed further simulations assuming a lower reduction on transmissibility due to the process of immunization, respectively for 80% and 65%. We added these results in the SI, see new Figs. S12-S13. In the case of lower immunization efficacy, the epidemic evolves faster and the susceptible pool represented by workers depletes faster. Passengers keep changing every day, hence their number of infections will depend on the number of active cases among workers in the airport day by day. Note that recent experiments applying UV-C lamps showed that a reduction of 93-98% of the pathogen load of airborne pathogens can be achieved in the context of a room. See Eadie, Ewan, et al. "Far-UVC (222 nm) efficiently inactivates an airborne pathogen in a room-sized chamber." *Scientific reports* 12 (2022): 4373. (<https://www.nature.com/articles/s41598-022-08462-z#Sec2>)

In practice, how would an airport even know of an outbreak occurring within their walls so early on? The scenarios presented here look at the results of immunizing within the first few days of the outbreak. Have you considered scenarios where outbreak identification happens later on? How quickly does the airport need to identify the outbreak among, for example their staff who might have reasons to not report symptoms? What is the efficacy of these immunization strategies if not everyone reports when they are infected?

We thank the reviewer for pointing this out, this is certainly the main point of our work. In our scenario the airport health authorities are not aware at all that an outbreak developed in the area. This is why we simulate all diseases spreading in absence of any other protocols, like testing-isolation, mask mandates, preventive quarantine of case contacts. In the case of outbreak identification, further protocols for outbreak control would come into play, joining hands with our spatial immunization policy. In brief, our method plays as a silent background protocol of outbreak prevention active at all times that should be installed in the first place. The importance of this method relies on its ability to reduce outbreak probability and intensity in case of importation of the first cases of completely undetected emerging diseases. The number of areas where to

intervene is strongly limited, thanks to the spatial heterogeneity of contact density, this is what makes our method feasible in practice. The under-reporting and loss of adherence to protocols would surely affect the efficacy of other protocols like test-isolate protocols, preventive quarantines of case contacts and mask mandates, but they do not affect the efficacy of our spatial immunization, since its efficacy does not depend on detection. The same modeling framework with further complications to include responses can be applied in later stages of the propagation, but this goes beyond the scope of this first work. We added this clarification for the scope of our work in the last paragraph of the introduction.

Beyond this, I think a check for grammar and spelling would benefit the article's readability, but overall I find the article is clearly laid out and interesting to read, even if I'm not sure that it would be a feasible approach for airports in practice.

We thank the reviewer for the positive opinion. We have revised grammar and spelling again.

Note that in recent experiments applying UV-C lamps, researchers showed that a reduction of 93-98% of the pathogen load of airborne pathogens can be achieved in the context of a realistically sized room. See Eadie, Ewan, et al. "Far-UVC (222 nm) efficiently inactivates an airborne pathogen in a room-sized chamber." *Scientific reports* 12.1 (2022): 4373.

(<https://www.nature.com/articles/s41598-022-08462-z#Sec2>)

We hope that the new version of our manuscript will now convince the reviewer of this method's importance and feasibility and meet the requirements for acceptance.

REVIEWERS' COMMENTS

Reviewer #1 (Remarks to the Author):

Thanks for the reviewed manuscript.
I particularly appreciate that the authors did re-run their simulation with an infectious period which follows a Gamma-distribution. In my opinion, the authors did respond to all questions in an adequate way.

Reviewer #2 (Remarks to the Author):

The authors have addressed all my comments and the new version of the manuscript has been improved.

Reviewer #3 (Remarks to the Author):

I thank the authors for taking my comments into consideration and for clarifying some of the article's points with additional writing and simulations. I appreciate the clarification of the UV approach being a silent tool running in the background so that constant monitoring of cases may not be required. At this point I think the article is good for publication.